# Agreement between In-Clinics and Virus Neutralization Tests in Detecting Antibodies against Canine Distemper Virus (CDV)

**DOI:** 10.3390/v14030517

**Published:** 2022-03-03

**Authors:** Sara Meazzi, Joel Filipe, Alessandra Fiore, Santina Di Bella, Francesco Mira, Paola Dall’Ara

**Affiliations:** 1Department of Veterinary Medicine and Animal Sciences, University of Milan, 26900 Lodi, Italy; joel.soares@unimi.it (J.F.); alessandra.fiore93@gmail.com (A.F.); paola.dallara@unimi.it (P.D.); 2Istituto Zooprofilattico Sperimentale della Sicilia “A. Mirri”, 90129 Palermo, Italy; santina.dibella@izssicilia.it (S.D.B.); dottoremira@gmail.com (F.M.)

**Keywords:** virus neutralization, VacciCheck, canine distemper virus, antibody titer, vaccination

## Abstract

Core vaccinations and specific antibody titer evaluations are strongly recommended worldwide by all the vaccination guidelines. Virus neutralization (VN) is considered the gold standard for measuring antibody titer against canine distemper virus, but it is complex and time consuming, and the use of in-clinics tests would allow to obtain quicker results. The aim of this study was to evaluate the agreement of the commercial in-clinics VacciCheck test compared to VN. A total of 106 canine sera were analyzed using both methods. The best agreement was obtained using a protective threshold of ≥1:32. VacciCheck showed 95.5% sensitivity, 87.2% specificity, and 92.5% accuracy. The Cohen’s kappa coefficient between methods was 0.84 (CI 95% 0.73 to 0.95), revealing an optimal agreement between the two methods (*p* = 0.0073). The evaluation of discordant results reveal that most samples had less than 1.5 dilution difference, and that usually did not affect the classification as protected or non-protected. Results also suggest that, in dubious cases, especially when a protective result is expected, retesting is advisable. In conclusion, VacciCheck may be considered as a reliable instrument that may help the clinician in identifying the best vaccine protocol, avoiding unnecessary vaccination, and thus reducing the incidence of adverse effects.

## 1. Introduction

Canine distemper is a severe infectious disease of the dog. Despite the wide host range, including many wild species belonging the families of *Canidae*, *Mustelidae*, *Procyonidae*, *Ursidae,* and *Viverridae*, dogs represent the main reservoir for canine distemper virus (CDV) [1]. In Italy, distemper infection had been reported in different southern areas due to the issue of stray dogs’ circulation, but it represents an emerging problem in the whole country due to the trade of importation of illegal puppies from east Europe [2,3]. It is caused by CDV, an enveloped, single negative-stranded RNA virus belonging to the genus *Morbillivirus*, family *Paramyxoviridae*, which also includes measles virus (MeV), porcine distemper virus (PDV), peste des petits ruminants virus (PPRV), rinderpest virus (RPV), and cetacean morbillivirus (CeMV). CDV is a highly contagious pathogen with mortality rates up to 80% and lack of an effective antiviral treatment [4]. Dogs could be infected at any age, but puppies between 6 weeks and 6 months of age are particularly prone to contract the infection due to the progressive decrease of maternally derived antibodies (MDA) [1]. For this reason, vaccination is considered the main method to control the disease and to reduce the severity of clinical signs. Indeed, all the international guidelines for the vaccination of dogs (World Small Animal Veterinary Association—WSAVA, American Animal Hospital Association—AAHA, Australian Veterinary Association—AVA, British Small Animal Veterinary Association—BSAVA, Canadian Veterinary Medical Association—CVMA) recommend that all dogs should be vaccinated with core vaccines against viruses causing parvovirus infection (CPV-2), canine distemper (CDV), and infectious canine hepatitis (CAdV-1) at least once in their life both to prevent individual infections and to assure herd immunity, thus reducing the prevalence of these threatening diseases [5,6,7,8,9]. However, several factors can interfere with the mount of an adequate immune protection, particularly against CDV. Firstly, CDV proliferate into lymphoid organs, causing leukopenia and lymphopenia, thus with a possible immunosuppressive effect [4]. Moreover, there may be an interference with high titers of MDA: usually, these antibodies protect puppies up to 9–12 weeks of age [1,2], but the duration of this passive protection is complicated by a high individual variability in the timing of MDA decline based on dams’ vaccination status, magnitude of colostrum intake, and environmental infective pressure [10,11]. To overcome this major problem, administering the initial core vaccinations in puppies at 6–8 weeks of age, then every 3–4 weeks until 16 weeks of age or older is recommended by all the vaccination guidelines and experts [2,5,6,7,8,9]. However, the exact knowledge of serum MDA and its interference on vaccination response as well as the puppy’s antibody protection status would have many positive implications; it would reduce (i) interferences in vaccination response, (ii) vaccination failures, and (iii) unnecessary vaccinations that could be associated with adverse reactions [12,13,14]. Moreover, antibody titration could allow to identify the so-called non-responder dogs, meaning those dogs that are unable to mount a protective immunity after vaccination or pathogen direct contact (generally they fail to seroconvert to one of the core vaccine antigens, such as CPV-2 or CDV). In literature, it is reported that 1 out of 5000 dogs may be a non-responder to CDV [2,15]. For all these reasons, in recent years, serological testing has been progressively introduced in veterinary practices to know the real protection status of dogs. 

The gold standard for detection of CDV antibodies in dogs is the virus neutralization (VN) test that is usually performed in specialized diagnostic laboratories [15]. However, the WSAVA guidelines supports the use of rapid serological in-clinic tests for the determination of antibody titers for core vaccines in dogs [5]. One of those tests is VacciCheck, which could be used to evaluate the immunoglobulin G (IgG) antibody titers against viruses causing parvovirus infection (CPV-2), canine distemper (CDV), and infectious canine hepatitis (CAdV-1) [16,17]. 

Thus, the aim of this study was to assess the performances of the in-clinics canine VacciCheck compared to those of the virus neutralization, which is considered the gold standard for the evaluation of antibody titer against CDV.

## 2. Materials and Methods

### 2.1. Study Population and Study Protocol

Serum samples used for this study were retrospectively selected from the database of VacciCheck results performed at the Department of Veterinary Medicine and Animal Sciences of the University of Milan, Italy. The only inclusion criterion was the availability of 500 µL of leftover serum, stored at −20 °C, to be sent to the Istituto Zooprofilattico Sperimentale della Sicilia “A. Mirri”, Palermo (Italy), where the virus neutralization test (VN) to assess humoral immunity against CDV was performed. According to the Ethical Committee decision of the University of Milan, residual aliquots of samples or tissues collected under informed consent of the owners can be used for research purposes without any additional formal request of authorization (EC decision 29 October 2012, renewed with the protocol n° 02-2016).

At first, to calculate sensitivity, specificity, and accuracy of VacciCheck, results were considered positive if the antibody titer was ≥1:32, following the manufacturer instruction (Biogal, Kibbutz Galed, Israel, see later). Conversely, based on literature, for the purposes of this study, virus neutralization was considered as protective when antibody titer resulted ≥1:16 [18,19]. Then, the analysis was repeated using the same threshold of positivity (≥1:32 or ≥1:16) for both methods to reduce possible bias. Finally, the level of agreement with the gold standard was calculated, dividing all the samples into four categories (negative, low positive, medium positive, and high positive) as reported in Table 1 (the positivity thresholds were different for the two methods). Then, the categories division was recalculated according to the VacciCheck column when using a positivity threshold of 1:32 for both methods and according to the VN column when using a positivity threshold of 1:16 for both methods.

### 2.2. Detection of CDV Antibodies by VacciCheck

VacciCheck (simplified name of ImmunoComb (IMB) VacciCheck^®^ Canine or ImmunoComb Canine VacciCheck^®^ Antibody Test Kit, produced in Israel by Biogal, Kibbutz Galed, Israel and supplied in Italy by Agrolabo, Scarmagno, Italy) relies on a semiquantitative ELISA method where the concentration of the antibodies is defined by the color intensity of the resulting spots compared with the “S” units on a scale from 1 to 6. A value of S3 (equal to an antibody titer of 1:32 for CDV) is considered a significant positive response by the manufacturer even if a S2 value (1:16) is also considered a weak positive. The whole scale (from 1 to 6) in S units correspond to antibody titers of <1:8, 1:16, 1:32, 1:64, 1:128, and 1:256, respectively. The manufacturer reported a sensitivity of 100% and a specificity of 92% for the detection of CDV antibodies [16]. 

### 2.3. Detection of CDV Antibodies by VN

In virus neutralization (VN) test, serum samples were inactivated in a water bath at 56 °C for 30 min, and two-fold dilutions of each serum sample (ranging from 1:2 to 1:216) were prepared in duplicate in Eagle’s Minimum Essential Medium (EMEM) (Sigma–Aldrich^®^, Milan, Italy) supplemented with an antibiotic and antimycotic solution (100 U/mL penicillin G sodium salt, 0.1 mg/mL streptomycin sulfate, and 0.25 μg/mL amphotericin B; EuroClone^®^, Milan, Italy) in a 96-well flat bottom plate. At each dilution well, 25 µL of 100 tissue culture infectious doses 50 (TCID50) of the Bussell variant of the Onderstepoort strain of CDV [20] was added. After incubation at 37 °C in a humid atmosphere of 5% CO_2_ for 1 h, 50 µL of a Vero cellular suspension (2 × 10^5^ cells/mL) maintained in EMEM supplemented with an antibiotic and antimycotic solution and 10% fetal calf serum (EuroClone^®^, Milan, Italy) was added to each serum/virus mixture. After 72 h incubation at 37 °C and 5% CO_2_ humidity, samples were observed using an inverted microscope at 50× magnification by specialized technicians that evaluated the presence of a cytopathic effect (CPE) in each well. The positive control serum was obtained from a previously tested, private-owned, vaccinated dog that showed a high antibody titer. A VN anti-CDV antibody titer equal to or above the first dilution (1:4) was considered positive. All the samples were performed in duplicate.

### 2.4. Statistical Analysis

Statistical analysis was performed using the Analyse-it software for Microsoft Excel. Specifically, sensitivity (Se) and specificity (Sp) of VacciCheck, both expressed as a percentage, were calculated according to the following formulas: Se = 100 × P/(TP + FN); Sp = 100 × TN/(FP + TN), where TP stand for true positive, FP for false positive, FN for false negative, and TN for true negative. The accuracy of VacciCheck, expressed as a percentage, was calculated according to the following formula: Acc = 100 × (TP + TN)/TP + FN + FP + TN).

The agreement was assessed using Cohen’s kappa coefficient both comparing only positive and negative results and dividing them in the four aforementioned categories (negative, low positive, medium positive, and high positive). Statistical significance was set at *p* < 0.05.

## 3. Results

### 3.1. Results of the Study Population

A total of 132 canine serum samples were first included in the study and sent to the Istituto Zooprofilattico Sperimentale della Sicilia for CDV virus neutralization assay. Because of insufficient volume or presence of some degree of cytotoxicity, 26 out of 132 samples were inadequate for performing the virus neutralization, and thus, they were excluded from further analysis. Hence, 106 serum samples were considered for statistical analysis. Results obtained from the in-clinics test VacciCheck and VN for CDV are reported in Table 2.

### 3.2. Agreement between Methods Using Positivity Threshold of 1:32 for VacchiCheck and of 1:16 for VN

Based on the positivity threshold of the tests (1:32 for VacciCheck and 1:16 for VN), the comparison of antibody titers resulted in 67 true positive (TP), 29 true negative (TN), 2 false positive (FP), and 8 false negative (FN). According to the obtained results, VacciCheck showed 89.3% sensitivity, 93.5% specificity, and 90.6% accuracy. The Cohen’s kappa coefficient was 0.78 (CI 95% 0.66 to 0.91) and revealed a good agreement with VN (*p* = 0.09). Dividing the results into positivity categories, the Cohen’s kappa coefficient was 0.65 (CI 95% 0.53 to 0.76), which revealed a moderate agreement with the viral neutralization test with the major discrepancies reported on medium- and high-positive results (*p* = 0.81).

### 3.3. Agreement between Methods Using a Positivity Threshold of 1:32

Using as positivity threshold an antibody titer of 1:32 for both methods, the comparison highlighted a difference in the total number of true positive (TP = 64), true negative (TN = 34), false positive (FP = 3), and false negative (FN = 5). In this case, VacciCheck showed 95.5% sensitivity, 87.2% specificity, and 92.5% accuracy. The Cohen’s kappa coefficient was 0.84 (CI 95% 0.73 to 0.95), which revealed an optimal significant agreement with the viral neutralization test (*p* = 0.0073). Dividing the results into positivity categories, the Cohen’s kappa coefficient was 0.63 (CI 95% 0.51 to 0.75), which revealed a good agreement with the viral neutralization test (*p* = 0.86).

### 3.4. Agreement between Methods Using a Positivity Threshold of 1:16

Using a positivity threshold of antibody titer of 1:16 for both methods, the comparison revealed 70 TP, 25 TN, 6 FP, and 5 FN results. In this case, VacciCheck showed 93.3% sensitivity, 80.6% specificity, and 89.6% accuracy. The Cohen’s kappa coefficient was 0.73 (CI 95% 0.61 to 0.89), which revealed a good agreement with the viral neutralization test (*p* = 0.26). Dividing the results into positivity categories, the Cohen’s kappa coefficient was 0.76 (CI 95% 0.66 to 0.86), which revealed a good agreement with the viral neutralization test (*p* = 0.13).

### 3.5. Discordant Results between Methods

The discordant results obtained using the three different thresholds are reported in Table 3, together with the antibody titers measured by the two different methods.

In the whole caseload, 21 samples had the same result for both VacciCheck and VN. For the remaining 85 samples, the majority (54 out of 85; 63.5%) showed a difference equal or less to one dilution. Eleven samples (12.9%) showed a difference of 1.5 dilutions. All the 14 samples (16.5%) that showed a difference of two dilutions gave results in agreement with the gold standard (high positive or negative) no matter what the difference in the antibody titer. Six samples (7%) showed a difference of three or more dilutions.

## 4. Discussion

A modern and evidence-based medical approach requires veterinarians to rely on a tailored vaccination program. Thus, the evaluation of antibody titer for CDV would allow to perform vaccination only when strictly necessary and reduce not only the chance for vaccination failures or adverse effects but also the possible spread of CDV infection [1]. However, the use of the gold-standard method, namely virus neutralization (VN), for the measurement of antibody titers would be, above all, time consuming since the samples have to be shipped to specific diagnostic laboratories. Thus, in the last years, the use of in-clinics serological methods, which are easy and rapid to perform, have been suggested even by WSAVA guidelines [5]. Among all the in-clinics tests available for the measurement of vaccinal protection in dogs and cats, VacciCheck is commercially available worldwide, and it is the only kit on the market approved for use by regulatory authorities for example in USA (USDA), Canada (CFIA), Japan (MAFF), and Brazil (ANVISA) [16]. To be reliable, VacciCheck results should agree with those of the gold standard (VN or hemagglutination-inhibition tests). In this study the aim was to investigate the agreement between VacciCheck and virus neutralization, which is considered as the gold standard for the evaluation of antibody titers against CDV. Results highlighted how the use of a positivity threshold of 1:32 for both the methods allow to maximize sensitivity and specificity of canine VacciCheck for CDV. Indeed, in this case, the result of sensitivity and specificity are closer albeit a little lower than those reported by the VacciCheck manufacturer (96% vs. 100% sensitivity and 87% vs. 92% specificity).

Most of the discordant results (in terms of false positive or false negative) had only one dilution difference, and this may rely on some possible misclassification in the evaluation of the VacciCheck colorimetric scale or for the complexity of the VN reading. It should be noted that, only in 6.6% of cases (7 out of 106), VacciCheck resulted negative, while VN resulted protective. Contrarily, in 1.9% of cases (2 out of 106) VN resulted non-protective, while VacciCheck resulted protective. In all those cases, the dilution difference was 1.5 or less. Thus, in those animals, the clinicians may suggest to repeat the test or to perform VN, especially in those breeds in which vaccine-associated adverse events (VAAEs) are reported with a higher frequency (i.e., small or toy breeds) [21]. This may have interfered with the calculation of sensitivity, specificity, and accuracy as well, especially in those few cases in which a one-dilution difference would discriminate a protective from an unprotective immunity.

Among the discordant results, those with at least three dilutions of difference were the most challenging to interpret. Specifically, three dogs showed a VacciCheck titer of 1:64 vs. a VN titer of 1:512 (ID 27, 38, and 72). In those cases, the first two dogs received the last vaccination less than one year before the sampling, while for the third one, there was no information regarding vaccination status. Nevertheless, this difference in the antibody titer would not influence the classification of those dogs as protected against CDV infection. Another four samples had a VacciCheck titer of 1:16, while VN was negative (ID 59, 65, 68, and 103). Of those animals, four were puppies younger than six months of age, while the last one was an adult dog without information about the vaccination status. Again, using a positivity threshold of 1:32, in all cases, the dogs would be considered as not protected for CDV infection. Another case (ID 70) without any vaccinal information had 3.5 dilution of disagreement (VacciCheck 1:16 vs. 1:192 VN), and in this case, the difference with VN would cause a misclassification of the dog as unprotected. The last two samples with four dilutions of disagreement were obtained from one dog (ID 16) with the last vaccination performed less than two years before the sampling and with clinical signs of *Leishmania* spp. infection. The other one (ID 28) was a geriatric dog vaccinated less than one year before the sampling. Thus, in case of negative results, especially when a positive result is expected (e.g., puppies at the end of first vaccinal cycle or adults after 1–2 years from the last core vaccinations), if results are low, positive or negative sample retesting or VN performing is advisable. Moreover, it should be remembered that VN, although sensitive and specific, is also a complex and time-consuming assay that requires cell culture and live virus manipulation; thus, it is plausible that there may be some differences (and probably less discrepancies) if the test was repeated. 

The results of this study highlight a slightly lower value of sensitivity and specificity compared with those reported in the experimental trial performed in Professor Schultz’s Wisconsin lab (those reported in the leaflet of VacciCheck) [22]. A possible explanation of this difference may rely on the different number of samples used in our study (since 26 samples had to be excluded due to preanalytical factors). Moreover, in the performance study by Schulz, there is a reduced number of dogs with negative antibody titers, and this may have influenced their results of accuracy and specificity. Another interesting point concerns the possible cross-reaction of other *Morbilliviruses*. Even though CDV is the only canine-specific *Morbillivirus*, recently, a new *Morbillivirus* that usually infect cats (namely feline morbillivirus-1, FeMV-1) was rarely isolated from dogs that present with respiratory diseases [23]. This virus has been isolated also in cats in Northern Italy [24]. Thus, Italian dogs may be exposed to FeMV-1 even though there is a lack of prevalence study in literature. However, this may be beneficial due to the cross-protection generally induced by *Morbilliviruses* infection [25]. One example of the *Morbillivirus* cross-protection is the use of human measles vaccine for protecting puppies against CDV in order to bypass the MDA interference [2]. The evaluation of possible cross-protection was beyond the aim of this study, but this may be an interesting topic for future studies.

## 5. Conclusions

In conclusion, as suggested by all vaccination guidelines and experts [5,6,7,8,9], vaccinations should be considered as one component of a comprehensive preventive healthcare plan that should be tailored to individual features (e.g., age, breed, health status, environment, lifestyle, etc.). This would represent the best evidence-based approach to vaccination. To avoid a blind vaccination against core diseases (caused by CDV, CAV, and CPV-2), veterinarians should evaluate the real need of a booster in each patient, and this may be supported by the use of in-clinics rapid test. The in-clinics canine VacciCheck showed a good agreement with VN, demonstrating its reliability in the evaluation of antibody titers against CDV. Thus, it may be considered as an efficient tool in daily practice, helping clinicians in following the evidence-based vaccinal approach and evaluating case-by-case the real need of a re-vaccination against canine distemper.

## Figures and Tables

**Table 1 viruses-14-00517-t001:** Categorization based on antibody titer for VacciCheck (positivity threshold ≥1:32) and virus neutralization (VN) (positivity threshold ≥1:16).

Categories	VacciCheck	Virus Neutralization (VN)
Negative	<1:32	<1:16
Low Positive	≥1:32–1:64	≥1:16–1:32
Medium Positive	≥1:64–1:128	≥1:32–1:64
High Positive	≥1:128	≥1:64

**Table 2 viruses-14-00517-t002:** Results obtained for each sample for both VacciCheck and VN.

ID	VacciCheck	VN	Titer Difference
1	64	64	-
2	64	128	1
3	32	16	1
4	16	24	0.5
5	128	512	2
6	64	96	0.5
7	256	>512	1
8	32	16	1
9	8	24	1.5
10	<8/NEGATIVE	8	1
11	256	>512	1
12	64	64	-
13	256	>512	1
14	32	64	1
15	128	64	1
16	8	128	4
17	128	>512	2
18	128	512	2
19	256	512	1
20	64	192	1.5
21	256	>512	1
22	32	48	0.5
23	64	128	1
24	8	24	1.5
25	256	>512	1
26	64	192	1.5
27	64	>512	3
28	128	8	4
29	128	64	1
30	128	96	0.5
31	128	256	1
32	128	384	1.5
33	128	>512	2
34	128	>512	2
35	64	192	1.5
36	64	256	2
37	128	96	0.5
38	64	>512	3
39	128	>512	2
40	128	64	1
41	128	192	0.5
42	128	48	1.5
43	>256	>512	1
44	64	128	1
45	256	>512	1
46	>256	>512	1
47	>256	512	1
48	256	>512	1
49	128	128	-
50	>256	96	1.5
51	>256	>512	1
52	>256	>512	1
53	>256	192	0.5
54	>256	384	0.5
55	>256	512	1
56	256	>512	1
57	<8	24	1.5
58	128	96	0.5
59	16	<8/NEGATIVE	2
60	>256	384	0.5
61	16	32	1
62	>256	>512	1
63	<8/NEGATIVE	<8/NEGATIVE	-
64	256	>512	1
65	16	<8/NEGATIVE	2
66	128	96	0.5
67	128	>512	2
68	16	<8/NEGATIVE	2
69	32	64	1
70	16	192	3.5
71	8	<8/NEGATIVE	1
72	64	512	3
73	>256	>512	1
74	<8/NEGATIVE	<8/NEGATIVE	-
75	<8/NEGATIVE	<8/NEGATIVE	-
76	>256	>512	1
77	128	128	-
78	<8 or NEGATIVE	<8/NEGATIVE	-
79	<8/NEGATIVE	<8/NEGATIVE	-
80	>256	>512	1
81	<8 or NEGATIVE	16	2
82	256	>512	1
83	256	>512	1
84	<8 or NEGATIVE	<8/NEGATIVE	-
85	>256	512	1
86	8	<8/NEGATIVE	1
87	8	<8/NEGATIVE	1
88	8	<8/NEGATIVE	1
89	8	<8/NEGATIVE	1
90	<8/NEGATIVE	<8/NEGATIVE	-
91	8	<8/NEGATIVE	1
92	<8/NEGATIVE	<8/NEGATIVE	-
93	<8/NEGATIVE	<8/NEGATIVE	-
94	<8/NEGATIVE	<8/NEGATIVE	-
95	>256	384	1.5
96	<8/NEGATIVE	<8/NEGATIVE	-
97	64	16	2
98	<8/NEGATIVE	<8/NEGATIVE	-
99	8	6	0.5
100	<8/NEGATIVE	<8/NEGATIVE	-
101	32	12	1.5
102	<8/NEGATIVE	<8/NEGATIVE	-
103	16	<8/NEGATIVE	2
104	<8/NEGATIVE	<8/NEGATIVE	-
105	<8/NEGATIVE	<8/NEGATIVE	-
106	<8/NEGATIVE	<8/NEGATIVE	-

- = no titer difference.

**Table 3 viruses-14-00517-t003:** Discordant result (false-positive and false-negative VacciCheck) compared to the gold standard (virus neutralization). For each sample are reported the antibody titer measured by VacciCheck (first titer) and VN (second titer). In bold are samples that resulted discordant for all the three thresholds examined.

	Positivity Threshold		
	VacciCheck 1:32 vs. VN 1:16	Both 1:32	Both 1:16
FP	**ID28 =** 1:128–1:8	ID3 = 1:32–1:16	**ID28****=** 1:128–1:8
	**ID101****=** 1:32–1:12	ID8 = 1:32–1:16	ID59 = 1:16–1:8
		**ID28 =** 1:128–1:8	ID65 = 1:16–1:8
		ID97 = 1:64–1:16	ID68 = 1:16–1:8
		**ID101****=** 1:32–1:12	**ID101****=** 1:32–1:12
			ID103 = 1:16–1:8
FN	ID4 = 1:16–1:24	**ID16****=** 1:8–1:128	ID9 = 1:8–1:24
	ID9 = 1:8–1:24	ID61 = 1:16–1:32	**ID16****=** 1:8–1:128
	**ID16****=** 1:8–1:128	ID70 = 1:16–1:192	ID24 = 1:8–1:24
	ID24 = 1:8–1:24		ID57 = 1:8–1:24
	ID57 = 1:8–1:24		ID81 = 1:8–1:16
	ID61 = 1:16–1:32		
	ID70 = 1:16–1:192		
	ID81 = 1:8–1:16		

## Data Availability

The authors confirm that the datasets analyzed during the study are available from the first author or the corresponding author upon reasonable request.

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
