# Peer review of "Agreement between In-Clinics and Virus Neutralization Tests in Detecting Antibodies against Canine Distemper Virus (CDV)"

_viruses, 2022, doi:10.3390/v14030517_

Round 1
Reviewer 1 Report
Thank you for this paper seeking evidence to support the use of a commercial test for CDV antibodies for use in general practice to provide more assurance to clients of the efficacy of the procedure and protection achieved. The principle is fine but the approach is a little unusual. Has the test method been approved for ELISA and CDV by OIE? If it has been independently validated methodologically then all you have to do with clients is ensure that they understand the sensitivity and specificity - or chance of false positives (potentially harmful) and negatives (re-vaccination will not likely do any harm). The attempt to show that the test works by doing a kappa comparison on results with a gold standard VN seems effectively redundant in this case? This is a random sera set and to get valid comparable results you need a set of sera with known exposure negative, exposure positive and vaccinated positive and negative then look at the results. Perhaps if you can provide a clearer picture of what the sera set actually represents - all vaccinated? Non responsive included? You do not provide the likely bias in this set and how this may confound the results. You mention the size of the set and again we should have some statistical figure as to how many samples are necessary to deal with the high sensitivity and lack of specificity in the test and come up with a reasonable result. What happens if the dog was exposed to another morbillivirus? Unlikely but still you need to discuss this sort of issue. So more information on the limitations of the test. Finally it would be sensible if you provide some recommendations on how you handle results - positives ok but negatives or low positives? Do you send for VN or do you revaccinate. Looks a simple story but never is with serology. I have made a few points in the text.

Author Response
Thank you for this paper seeking evidence to support the use of a commercial test for CDV antibodies for use in general practice to provide more assurance to clients of the efficacy of the procedure and protection achieved. The principle is fine but the approach is a little unusual.
Has the test method been approved for ELISA and CDV by OIE?
Author’s answer: The test method compared in this study has not been approved by OIE but it followed a standardized validation method reported by the manufacturer (it should be remember that Israeli is an OIE member). Data on the value of this semiquantitative in-clinic ELISA test for determining CDV and CPV antibody levels in dogs before annual revaccination were thus provided (Waner et al., J Vet Diagn Invest. 2006 May;18(3):267-70 + Schultz et al). Similarly, another in-clinic commercial ELISA test kit for the determination of protective serum antibody concentrations in dogs against CPV and CDV has similar applications since test results are correlated with the gold-standard assays (Carmichael, J Vet Med B Infect Dis Vet Public Health. 2005 Sep-Oct;52(7-8):303-11; Litster et al., Vet J. 2012 Aug;193(2):363-6.). Moreover, another study using virus neutralization as the reference standard was recently performed to evaluate and compare the diagnostic accuracy of four point-of-care tests, also including the above cited ELISA tests (Bergmann et al., Vet J. 2021 Jul;273:105693).
Micheal J. Day – Emeritus Professor of Clinical Immunopathology, School of Veterinary Sciences, University of Bristol (UK) and the main extensor of the WSAVA (World Small Animal Veterinary Association) vaccination guidelines for dogs and cats – at the 39th WSAVA Congress (2014) suggested the use of both those ELISA test (namely VacciCheck and TiterChek) as a well validated tests compared to the ‘gold standard’ VN (virus neutralization) and HI (haemagglutination inhibition) tests [https://vaccicheck.com/wp-content/uploads/2012/04/WSAVA2014.pdf]. Indeed, measurement of CDV and CPV serum antibody titers may be a useful tool for determining the need for vaccination before annual revaccination and CDV titers are commonly measured using ELISA, indirect fluorescent antibody assay and serum neutralization tests (Tizard & Ni, J Am Vet Med Assoc. 1998 Jul 1;213(1):54-60).
As confirmed also by Biogal (the Israeli manufacturer of the test), VacciCheck is nowadays the only commercial kit available that is approved for the three core diseases (CDV, CPV, CAV) by regulatory authorities in many countries, including the USDA (United States Department of Agriculture)/FDA (Food and Drug Administration) in USA, the CFIA (Canadian Food Inspection Agency) in Canada, the MAFF (Ministry of Agriculture, Forestry and Fisheries) in Japan, the ANVISA (Agência Nacional de Vigilância Sanitária) in Brazil (https://www.biogal.com/products/vaccicheck/), and in many other countries.
We included in the manuscript and in the reference list the above cited studies to further underline the validity of the ELISA method for its clinical purpose to underline the aim and clarify the approach of this study.
If it has been independently validated methodologically then all you have to do with clients is ensure that they understand the sensitivity and specificity - or chance of false positives (potentially harmful) and negatives (re-vaccination will not likely do any harm).
Author’s answer: We agree with the reviewer that this is one of the veterinarian duties in his daily practice: make sure the owners fully understand these important aspects. For the same reason, an independent comparison between the ELISA and virus neutralization tests, performed on of clinical routine sera collected from dogs under field conditions, could further clarify any limits of the test itself.
The attempt to show that the test works by doing a kappa comparison on results with a gold standard VN seems effectively redundant in this case?
Author’s answer: According to the paper of Watson and Petrie [Method agreement analysis: A review of correct methodology. Theriogenology 73 (2010) 1167–1179] Cohen’s kappa is suggested to provide a measure of agreement between methods when the results are categoric variable and it is useful to investigate if a quicker, cheaper, or otherwise more efficient test may replace the gold-standard. For this reason, we used the Cohen’s Kappa. Similarly, the same statistic coefficient (κ) was used also in the compared study (Bergmann et al., Vet J. 2021 Jul;273:105693) to measure inter-rater reliability. Some clinicians remain doubtful about the effectiveness and reliability of VacciCheck, since there is a lack of independent agreement studies between VacciCheck and VN for CDV (whereas there are some paper for CPV in-clinics kits and the gold standard HI) in literature.
This is a random sera set and to get valid comparable results you need a set of sera with known exposure negative, exposure positive and vaccinated positive and negative then look at the results. Perhaps if you can provide a clearer picture of what the sera set actually represents - all vaccinated? Non responsive included? You do not provide the likely bias in this set and how this may confound the results.
Author’s answer: All the sera used in this study belong to owned and kennel dogs routinely tested with VacciCheck in the daily practice to know their protective status. The caseload includes dogs of different ages (puppies, adults, old), breeds (purebred or crossbred), sizes (small, medium, large), vaccination status (vaccinated or not vaccinated) and health status. Since they are part of the routine examination to assess their protection status against distemper, we aimed to demonstrate the good agreement with the gold standard test. If the reviewer think that is information may be important for the manuscript we can add a table as supplementary data.
You mention the size of the set and again we should have some statistical figure as to how many samples are necessary to deal with the high sensitivity and lack of specificity in the test and come up with a reasonable result.
Author’s answer: There is no real information about the prevalence of CDV protection in canine population, thus a precise sample number could not be decided before the experimentation. However, we assumed that the majority of the tested population would be protected and since both tests presented high specificity and sensitivity, we assumed that a number of at least 100 samples would be enough to confirm our hypothesis of good agreement between methods. Indeed, only very few samples were discordant in defining the protective status, thus confirming our theory and demonstrating the good agreement between VacciCheck and the gold standard.
What happens if the dog was exposed to another morbillivirus? Unlikely but still you need to discuss this sort of issue.
Author’s answer: Thank you very much for this intriguing question. As far as we know, Canine Distemper Virus (CDV) is the only morbillivirus specific for the canine species. However, recently a new morbillivirus that occurs in cats (namely Feline morbillivirus-1, FeMV-1) is reported to be rarely isolated from infected dogs that present with respiratory diseases [Piewbang C., Wardhani S.W., Dankaona W., Yostawonkul J., Boonrungsiman S., Surachetpong S., Kasantikul T., Techangamsuwan S. (2021): Felinemorbillivirus-1 in dogs with respiratory diseases. Transboundary and Emerging Diseases, 1-10]. This virus has been isolated also in Northern Italy cats [Stranieri A., Lauzi S., Dallari A., Gelain M.E., Bonsembiante F., Ferro S., Paltrinieri S. (2019): Feline morbillivirus in Northern Italy: prevalence in urine and kidneys with and without renal disease. Veterinary Microbiology, 233, 133-139]. Thus, the exposition of Italian dogs to FeMV-1 is possible, even though there is a lack of prevalence study in literature. However, this may be beneficial, due to the cross-protection generally induced by Morbilliviruses infection [de Vries R.D., Duprex W.P., de Swart R.L. (2015): Morbillivirus infections: an introduction. Viruses, 7, 699-706]. One example of the morbillivirus cross-protection is the use of human measles vaccine for protecting puppies against CDV in order to bypass the MDA interference [8]. We have added this information in the Discussion section of the manuscript.
So more information on the limitations of the test.
Author’s answer: Based on our experience of several years, this test has really few limitations. It is rapid and cheap, with high accuracy and reproducibility. Moreover, it needs a very little amount of blood volume (5-10 mL) and it can be helpful for clinician in order to choose a tailored vaccinal approach for each patient. The main limitation is the test specificity that do not reach 100%, but this point has already been discussed and it does not represent a main issue in the clinical practice. Is should be noted that both VacciCheck and VN rely on operator-dependent (meaning subjective) results reading, thus we think that this cannot be considered as a limitation compared to the gold standard.
Finally it would be sensible if you provide some recommendations on how you handle results - positives ok but negatives or low positives? Do you send for VN or do you revaccinate. Looks a simple story but never is with serology.
Author’s answer: Thank you for your feedback. Following your suggestion, the last sentence has been modified as follows: As suggested by all vaccination guidelines and experts [3-7], vaccinations should be considered as one component of a comprehensive preventive health care plan that should be tailored on individual features (e.g. age, breed, health status, environment, lifestyle, etc.). This would represent the best evidence-based approach to vaccination. To avoid a blindly vaccination against core diseases (caused by CDV, CAV, and CPV-2), veterinarians should evaluate the real need of a booster in each patient and this may be supported by the use of in-clinics rapid test. The in-clinics canine VacciCheck showed a good agreement with VN, demonstrating its reliability in the evaluation of antibody titers against CDV. Thus, it may be considered as an efficient tool in the daily practice, helping clinician in following the evidence-based vaccinal approach, evaluating case-by-case the real need of a re-vaccination against canine distemper.
Line 27: add “and many other wild species”
Author’s answer: Thank you for this correct suggestion. A sentence about the wild species has been added.
Line 28: specify “morbillivirus for the paromyxoviridae RNA etc..”
Author’s answer: Thank you for the suggestion. All these information has been added in the manuscript.
Line 51 this sentence is unclear. Try to express what you mean more precisely
Author’s answer: In this sentence we mean that even if there is a vaccination scheme, the possibility to better know the pup’s exact vaccination response and how the MDA may influence this response would allow to perform vaccination only when needed. We try to modify the sentences, hopefully it is clearer now.
lines 55: presumably after one or more vaccinations?
Author’s answer: Thank you for this comment, we better specify this information in the manuscript.
Line 72: We need some reference to validation studies on the VACCICHECK what are the sensitivities and specificities reported in dog serum? What guarantee is there that the antibody in the sera is from CDV vaccine induced or natural, are there control sera known negatives and known natural positives?
Author’s answer: The main VacciCheck validation document is the reference 21: Biogal Galed Labs. Acs. Ltd. Performance report: A field and experimental trial to assess the performance of the ImmunoComb Canine VacciCheck Antibody Test Kit. 2015, 1–26. The study was conducted in 2015 in the Professor Schultz’s lab at the Department of Pathobiological Sciences, School of Veterinary Medicine, University of Wisconsin- Madison, USA (principal contact investigator: Laurie J. Larson) and is available online (https://vaccicheck.com/wp-content/uploads/2014/02/VacciCheck-Performance-Wisconsin.pdf). Furthermore, as reported in the first page of this document, VacciCheck has been officially approved by USDA, CFIA, MAFF and ANVISA.
Manufacturer’ Sensitivity (Se) and Specificity (Sp) are reported at lines 106 and 107: 100% Se and 92% Sp. as reported also at the end of the Discussion section. Our results highlight a slightly lower value of Se and Sp compared with those reported in the VacciCheck leaflet [Schultz and colleagues, 2015]. A possible explanation of this difference may rely on the reduced number of samples used in our study, but also on the subjective nature of VacciCheck and/or VN readings. Nevertheless, most of the discordant results had only one dilution difference that did not change the protected/unprotected status of the dog.
About vaccine or natural induced antibody, we do not think that this is relevant in this study, because the information needed (for an agreement evaluation) is the presence of absence of protective antibodies. Nevertheless, due to the low number on distemper cases in Northern Italy (where the dogs were enrolled), the chance to have a positive result due to a natural infection, instead of vaccination, is very low. Moreover, in a symptomatic dog VacciCheck, which measure the presence of IgG, may result negative, since the first antibody produced would be IgM (IgG would be produced in a second time, as “memory antibody”). Regarding the negative control sera, virtually all the unvaccinated dogs may tested negative due to the low prevalence of the field virus.
Line 81: this suggests you are validating these results but not sure you are following protocol for this according to OIE?
Author’s answer: Actually, a validation of the VacciCheck has already been performed. The aim of our study was to evaluate the agreement of VacciCheck with the gold standard (VN). This would allow a more feasible evaluation of the protection status that may be performed also by clinicians, avoiding the delay caused by the sample shipping to an external lab for the VN test evaluation.
Line 106: This means false positives are possible and false negatives.
Author’s answer: Most of the test used in the clinical practice do not reach 100% of sensitivity or specificity (or both). A sensitivity below 100% means the risks to provide false negative results, thus, in our case, some dogs that are protected (in most cases those with a low amount of IgG), may result negative with VacciCheck. Clinicians have two possibilities: re-test or vaccinate the dog. On the other hand, a specificity below 100% means the risks to provide false positive results, and, in this case, some dogs with positive VacciCheck result actually would not be protected against distemper (this is of course the worst scenario). Even though is important to keep in mind the characteristic of the test, it should be underlined that in many countries (Italy included) there is a lack of monovalent vaccines against distemper. Thus, in most cases, veterinarians had to use at least bivalent distemper-parvovirus vaccine, reducing the risk to have unprotected dogs even in case of false positive VacciCheck results for CDV.
Line 123: bit risky perhaps it was non responder?
Author’s answer: The positive control serum was obtained from a previously tested private-owned vaccinated dog, that showed a high antibody titer, thus it is unlikely to be a non-responder. We add this information in the paper.

Reviewer 2 Report
In this article, authors claim that the VacciCheck may be considered as a reliable instrument that may help the clinician in identifying the best vaccine protocol, avoiding unnecessary vaccinations.
How was the sample size of 132 canine serum samples concluded?
106 serum samples were considered for statistical analysis, is this a good sample size?
Can authors mention specifificty, sensitivity, recall and precision values for detections?
Are these titer measured in replicates?
Can authors provide data in supplementary section?
Titer differences are minimal but are they optimal?
Would appreciate authors responses on above. Thanks,
Author Response
In this article, authors claim that the VacciCheck may be considered as a reliable instrument that may help the clinician in identifying the best vaccine protocol, avoiding unnecessary vaccinations.
Would appreciate authors responses on above. Thanks,
How was the sample size of 132 canine serum samples concluded?
Author’s answer: Our “VacciCheck archive” include many yet tested frozen samples that have been used for several studies (some have been tested also with other techniques, e.g. HI for parvovirus infection or other in-clinics validation). For this study we had to choose the maximum samples number that had sufficient volume to perform also the VN (0.3 ml or more), in order to reach at least 100 samples, that could be considered a good sample for our purposes.
serum samples were considered for statistical analysis, is this a good sample size?
Author’s answer: There is no real information about the prevalence of CDV protection in canine population, thus a precise sample number could not be decided before the experimentation. However, we assumed that most of the tested population would be protected and since both tests presented high specificity and sensitivity, we assumed that a number of at least 100 samples would be enough to confirm our hypothesis of good agreement between methods. Indeed, only very few samples were discordant in defining the protective status, thus confirming our theory and demonstrating the good agreement between VacciCheck and the gold standard.
Can authors mention specifificty, sensitivity, recall and precision values for detections?
Author’s answer: We have some issue in understanding this question. If the reviewer means the value of specificity and sensitivity of VacciCheck those are reported at lines 218-219 (both those observed in this study and the ones reported by the manufacturer: 96% vs 100% sensitivity and 87% vs 92% specificity respectively).
Are these titer measured in replicates?
Author’s answer: All samples used for VN are performed in duplicate. Contrarily VacciCheck results have been included retrospectively and only in some cases were performed in duplicate, since they have been performed for diagnostic evaluation and not for research purposes (meaning that the owner had paid for it) . Nevertheless, all the VacciCheck samples performed in duplicate gave similar results.
Can authors provide data in supplementary section?
Author’s answer: What kind of supplementary data the reviewer think is needed?
Titer differences are minimal but are they optimal?
Author’s answer: As reported at the end of the Discussion section of the manuscript, our results highlight a slightly lower value of sensitivity and specificity compared with those reported in the leaflet of VacciCheck and obtained by Schultz and colleagues [21], probably due both to the reduced number of samples used in our study and to difficulties in VacciCheck and/or VN readings. In any case, most of the discordant results had only one dilution difference that did not change the protected/unprotected status of the dog. Thus, even if the agreement is not perfect (as it happens in most cases of agreement studies), the use of an in-clinics test such as VacciCheck would have a lot of advantages for clinicians, such as to avoid the delay of serum shipment to an external lab, to reduce cost of antibody titer evaluation and to perform vaccination only when needed. For these reasons, based on the results of this study, the use of VacciCheck in clinical practice is suggested.

Round 2
Reviewer 1 Report
Thank you for completing a thorough consideration of the review and for appropriate responses which I believe adequately address all my main concerns. You need to consider some minor issues on expression in the introduction which I highlight by comments on the PDF text. Otherwise I think good to go.

Author Response
Comments and Suggestions for Authors
Thank you for completing a thorough consideration of the review and for appropriate responses which I believe adequately address all my main concerns. You need to consider some minor issues on expression in the introduction which I highlight by comments on the PDF text. Otherwise I think good to go.
Line 29-32: this sentence reads clumsily with respect to the geographic association please revise both for Southern regioni and East Europe. Is East Europe a recognised geographic entity? Maybe use lower case east Europe which is a description used by WHO.
Author’s answer: Thank you for your suggestion. We modified the sentence as following: In Italy distemper infection had been reported in different southern areas due to the stray dog’s circulation issue, but it represents an emerging problem in the whole country due to the illegal puppies’ trade of importation from east Europe [2, 3].”
Line 36: and cetacean morbillivirus
Author’s answer: Thank you for noticing. We add this information following your suggestion.
